# Chitosan Covalently Functionalized with Peptides Mapped on Vitronectin and BMP-2 for Bone Tissue Engineering

**DOI:** 10.3390/nano11112784

**Published:** 2021-10-21

**Authors:** Paola Brun, Annj Zamuner, Leonardo Cassari, Gabriella D’Auria, Lucia Falcigno, Stefano Franchi, Giorgio Contini, Martina Marsotto, Chiara Battocchio, Giovanna Iucci, Monica Dettin

**Affiliations:** 1Department of Molecular Medicine, University of Padova, Via A. Gabelli 63, 35121 Padova, Italy; paola.brun.1@unipd.it; 2Department of Industrial Engineering, University of Padova, Via F. Marzolo 9, 35131 Padova, Italy; annj.zamuner@unipd.it (A.Z.); leonardo.cassari@phd.unipd.it (L.C.); 3L.i.f.e.L.a.b. Program, Consorzio per la Ricerca Sanitaria (CORIS), Veneto Region, 35128 Padova, Italy; 4Department of Pharmacy, University Federico II of Naples, Via Domenico Montesano 49, 80131 Naples, Italy; gabriella.dauria@unina.it (G.D.); falcigno@unina.it (L.F.); 5Istituto di Struttura Della Materia-CNR (ISM-CNR), Via Fosso del Cavaliere 100, 00133 Rome, Italy; stefano.franchi79@gmail.com (S.F.); giorgio.contini@ism.cnr.it (G.C.); 6Department of Physics, University of Rome Tor Vergata, Via Della Ricerca Scientifica 1, 00133 Rome, Italy; 7Department of Science, Roma Tre University of Rome, Via Della Vasca Navale 79, 00146 Rome, Italy; martina.marsotto@uniroma3.it (M.M.); chiara.battocchio@uniroma3.it (C.B.); giovanna.iucci@uniroma3.it (G.I.)

**Keywords:** chitosan, covalent grafting, osteoblasts, bioactive peptides, bone tissue engineering

## Abstract

Worldwide, over 20 million patients suffer from bone disorders annually. Bone scaffolds are designed to integrate into host tissue without causing adverse reactions. Recently, chitosan, an easily available natural polymer, has been considered a suitable scaffold for bone tissue growth as it is a biocompatible, biodegradable, and non-toxic material with antimicrobial activity and osteoinductive capacity. In this work, chitosan was covalently and selectively biofunctionalized with two suitably designed bioactive synthetic peptides: a Vitronectin sequence (HVP) and a BMP-2 peptide (GBMP1a). Nuclear magnetic resonance (NMR), X-ray photoelectron spectroscopy (XPS), and Fourier transform infrared spectroscopy (FT-IR) investigations highlighted the presence of the peptides grafted to chitosan (named Chit-HVP and Chit-GBMP1a). Chit-HVP and Chit-GBMP1a porous scaffolds promoted human osteoblasts adhesion, proliferation, calcium deposition, and gene expression of three crucial osteoblast proteins. In particular, Chit-HVP highly promoted adhesion and proliferation of osteoblasts, while Chit-GBMP1a guided cell differentiation towards osteoblastic phenotype.

## 1. Introduction

Nowadays, bone tissue engineering (BTE) still represents the most valid alternative way to try to overcome the actual limits in bone healing [1,2,3]. In fact, up to the present, the gold standard procedure followed by orthopedic surgeons for the reconstruction of massive bone defects is still represented by the use of autografts, allografts, or xenografts [4,5,6]. This procedure suffers from many limitations, such as morbidity in the donor site and limited availability for autografts [7,8], disease transmission, immunogenicity, and lack of osteogenic properties for allografts and xenografts [2,9,10].

A proper graft for bone-healing applications must possess adequate mechanical features and several biological properties, such osteogenesis, osteoinduction, osteoconduction, and osteointegration [2,11,12]. In order to overcome the actual grafts limitations, BTE has risen in the last three decades, proposing a series of combinations of scaffolding materials suitably enriched with bioactive molecules and cells [1]. The selection of an appropriate material for scaffold fabrication represents a critical stage in BTE [11,12,13], and the interest in using natural materials (e.g., collagen, silk, fibrin, glycosaminoglycans, hyaluronic acid, alginate, gelatin, etc.) is increasing [13,14,15,16]. In this scenario, chitosan, a deacetylated derivate of chitin, represents a widely used scaffolding material that has proved to be biocompatible, bioactive, biodegradable, and possesses antimicrobial properties [17,18,19,20,21]. Furthermore, chitosan is a very versatile material that can be used as film, hydrogel, 3D sponge, or nanofiber matrix [22]. In particular, 3D chitosan-based composite scaffolds are appropriate for BTE application due to their optimal open porosity and pore size (200–600 µm), which allow cell migration and a good exchange of nutrients and waste [22,23]. Due to the low Young’s modulus (around 6.8 ± 0.5 MPa) [13], lyophilized chitosan 3D scaffolds can find application in small bone defects, or can be used for long bone segmental defects in combination with other biomaterials that confer adequate mechanical properties, such as hydroxyapatite (HA) [24,25,26,27,28], tricalcium phosphate (TCP) [29], bioglass ceramic [30,31], alginate [32,33], and so on.

Unfortunately, several studies have suggested that chitosan’s healing potential is doubtable when used alone, but it can be increased simply combining it with other efficient biomaterials, growth factors, and cells [22]. In fact, it has been demonstrated that bone morphogenetic protein 2 (BMP-2), insulin-like growth factor 1 (IGF-1), vascular endothelial growth factor (VEGF), or BMP-7-loaded chitosan scaffolds are more effective in bone healing [34]. This is clearly understandable since osteointegration and the long-term success of an implant is strictly dependent on the complex reaction at the tissue–material interface, confirming the necessity of biochemical signals on the scaffold surface that permit an optimal interaction with the surrounding bio-environment [35].

Biochemical motifs can be conveyed to the implant site in different ways, as through inclusion in the material bulk or in a carrier, or they can be simply adsorbed or covalently linked to the surface of the biomaterial [36]. The biofunctionalization method is strictly dependent on the kind of bioactive motif needed. For example, pro-angiogenic sequences can be embedded in a carrier to control the kinetics of release, whilst in the case of adhesive sequences, they should be grafted to the implant surface to avoid adverse effects due to a premature release [37]. For this reason, the need to find an easy way to covalently anchor pro-adhesive stimuli onto a chitosan matrix in order to drive cellular response toward new bone generation and to accelerate bone healing is clear. In fact, cells are able to discriminate, via mechanosensoring, between adhesive motives belonging to physisorbed proteins and those firmly anchored on the material surface, preferring to establish a more stable adhesion structure with the latter [38].

In the literature, several methods have been used to covalently anchor bioactive motifs to chitosan, from the use of 1-ethyl-3-(3-dimethylaminopropyl)-carbodiimide (EDC) and N-hydroxy succinimide (NHS) [39,40], to the primary conversion of chitosan amine residue with N-(*m*-maleimidobenzoyloxy) succinimide (MBS) or introducing a 2-iminothiolane and then binding a cysteine-containing peptide [41,42], and others [43,44]. The functionalization chemistry here proposed consists of a reaction between an aldehyde group suitably introduced in the peptide sequence and chitosan amino residues, with a Shiff base formation afterwards reduced to amine [45,46]. One of the advantages of this strategy is represented by the possibility of anchoring a bioactive motif in a specific way through a single-step reaction, using aqueous solution and mild conditions. The Achilles heel of this strategy is represented by the chance to obtain a peptide carrying an aldehyde terminal in large quantities and at a high purity grade. With this in mind, besides the already proposed peptide synthesis route using a NovaSyn H-Phe-H resin, which allows obtaining a peptide with an aldehyde group in C-terminal position [45,46], an alternative route is proposed, which involves the insertion of a serine residue which then undergoes an oxidative cleavage to an aldehyde group [47].

The most widely used bioactive sequence for biomedical material functionalization is undoubtedly the Arginine-Glycine-Aspartic Acid (RGD) motif [48]. RGD is a short sequence belonging to fibronectin and laminin [48,49], two proteins of the extracellular matrix (ECM), and it’s a cell-binding peptide that has been demonstrated to enhance cell adhesion, growth, proliferation, and differentiation [50,51,52,53]. In the literature, many studies have demonstrated the benefits deriving from chitosan’s biofunctionalization with RGD [39,40,43,44]; nevertheless, in this work, a nonapeptide called HVP (FRHRNRKGY) deriving from human vitronectin is proposed, as, differently from RGD, it shows an osteoblast-selective adhesive mechanism involving proteoglycan-mediated interactions, thus HVP not only selects the attachment and supports differentiation of osteoblasts but also promotes cellular migration [54,55]. Furthermore, in order to guide cell differentiation towards an osteoblastic phenotype, a BMP-2 mimicking peptide named GBMP1a (PFPLADHLNSTNHAIVQTLVNSF) is covalently bound to chitosan. GBMP1a (60 ng/mL) significantly increased osteogenic differentiation of mesenchymal stem cells (+25%) in comparison with BMP-2 (100 ng/mL) [56]. In fact, the aim of this study is to propose a cheap and easy to obtain 3D-spongy chitosan scaffold suitably biofunctionalized for BTE, in a way that ensures faster bone healing by acting as an optimal support for osteoblasts migration, adhesion, proliferation, and differentiation.

## 2. Materials and Methods

### 2.1. Materials

Chitosan 70/1000 was purchased by Heppe Medical Chitosan GmbH (HMC^+^, Halle, Germany). All Fmoc protected amino acids, resins, coupling reagents 2-(1H-Benzotriazole-1-yl)-1,1,3,3-tetramethyluroniumhexafluorophosphate (HBTU), and Oxima Pure, used for peptide synthesis were from Novabiochem (Merck KGaA, Darmstadt, Germany). Piperidine, diethyl ether, N, N-diisopropylethylamine (DIPEA), trifluoroacetic acid (TFA), and dichloromethane (DCM) were from Biosolve (Leenderweg, Valkenswaard, The Netherlands). All other solvents such as N, N-dimethylformamide (DMF) and reagents as sodium (meta)periodate (NaIO_4_) were from Sigma-Aldrich (Merck KGaA, Darmstadt, Germany) unless otherwise indicated.

### 2.2. Peptide Synthesis and Characterization

#### 2.2.1. S-X-HVP

S-X-HVP (H-Ser-X-Phe-Arg-His-Arg-Asn-Arg-Lys-Gly-Tyr-NH_2_, where X represents 7-aminoheptanoic acid) was synthesized by standard Fmoc chemistry using rink amide resin (0.52 mmol/g; scale 0.125 mmoles) and a fully automated peptide synthesizer (Syro I, Multisynthec, Witten, Germany). The side chain protections employed were: Arg, Pbf; Ser, tBu; Lys, Boc, Asn and His, Trt. The Fmoc deprotection was carried out with two treatments with 40% and 20% piperidine/DMF for 3 min and 12 min, respectively. The resin loading with the first amino acid (Tyr) was carried out manually using 8-eq of Fmoc-Tyr-OH, 8-eq of PyOxim, and 16-eq of DIPEA for 1 h. The following two amino acids were coupled through a single coupling cycle. All coupling reactions were carried out using 5 eq of Fmoc protected amino acid and HBTU/Oxima Pure/DIPEA (5 eq of HBTU/Oxima pure and 10 eq of DPEA; 45 min) in DMF. All the remaining couplings were double. After the final Fmoc deprotection, the resin was washed with DCM and dried for 1 hr and 30 min under vacuum. The peptide was cleaved from the solid support with contemporary side-chain deprotection using the following mixture: 0.125 mL H_2_O MilliQ, 0.125 mL TES, 4.750 mL TFA (90 min, under magnetic stirring). After cleavage, the resin was filtered, the reaction mixture concentrated, and the crude peptide precipitated with cold ethyl ether.

The crude S-X-HVP (33 mg) was dissolved in 0.05% TFA in H_2_O MilliQ, and the solution was filtered and loaded on Nova-Pak HR C_18_ (6 µm, 60 Å, 7.8 mm × 300 mm, Waters) and separated in the following conditions: eluent A, 0.05% TFA in MilliQ water; eluent B, 0.05% TFA in CH_3_CN; gradient, from 0% B to 8% B in 2 min, then from 8% B to 20% B in 24 min; flow rate, 4 mL/min; detection at 214 nm. The final peptide was over 99% pure.

The chromatogram of purified S-X-HVP was obtained in the following conditions: column, Nova-Pak HR C_18_ (4 µm, 60 Å, 3.9 mm × 300 mm, Waters), injection volume, 35 µL of 1 mg/mL peptide solution; flow rate, 1 mL/min; eluent A, 0.05% TFA in water; eluent B, 0.05% TFA in CH_3_CN; gradient, from 13% B to 23% B in 20 min, detection at 214 nm. The retention time result was 9.187 min (Appendix A). Experimental mass: 1446.7 Da, theoretical mass: 1446.6 Da (4800 MALDI-TOF/TOF TM analyzer provided with 4000 Series Explorer TM software, Applied Biosystem/MDS Sciex, CA, USA) (Appendix A).

#### 2.2.2. GBMP1a

GBMP1a (H-Pro-Phe-Pro-Leu-Ala-Asp-His-Leu-Asn-Ser-Thr-Asn-His-Ala-Ile-Val-Gln-Thr-Leu-Val-Asn-Ser-Phe-CHO) was synthesized by standard Fmoc chemistry using H-Phe-H NovaSyn resin (0.18 mmol/g; scale 0.125 mmoles) and a fully automated peptide synthesizer (Syro I, Multisynthec, Witten, Germany). The side chain protections employed were: Ser and Thr, tBu; Asp, OtBu; Asn, Gln and His, Trt. The Fmoc deprotection was carried out with two treatments with 40% and 20% piperidine/DMF for 3 min and 12 min, respectively. All coupling reactions were carried out using 5 eq of Fmoc protected amino acid and HBTU/Oxima Pure/DIPEA (5-eq of HBTU/Oxima Pure and 10 eq of DIPEA; 45 min) in DMF. All couplings were double. After Fmoc deprotection, the resin was washed with DCM and dried for 1 hr under vacuum. Side-chain protecting groups were removed with a treatment with TFA for 1 hr, then the peptide was cleaved from the solid support using the following mixture: 0.5 mL H_2_O MilliQ, 2.1 mL methanol, 1 mL acetic acid and 6.3 mL DCM (1 hr, under magnetic stirring). After cleavage, the resin was filtered, the reaction mixture concentrated, and the crude peptide precipitated with cold ethyl ether. The obtained peptide GBMP1a already held an aldehyde group in C-terminal position.

The crude peptide (38 mg) was dissolved in H_2_O MilliQ, the solution was filtered and loaded on Jupiter C_18_ (5 µm, 300 Å, 10 mm × 250 mm, Phenomenex) and separated in the following conditions: eluent A, 0.05% TFA in MilliQ water; eluent B, 0.05% TFA in CH_3_CN; gradient, from 0% B to 25% B in 2 min, then from 25% B to 35% B in 40 min; flow rate, 4 mL/min; detection at 214 nm. The final peptide was 99.3% pure.

The chromatogram of purified GBMP1a was obtained using: column, Jupiter C_18_ (5 µm, 300 Å, 4.6 mm × 250 mm, Phenomenex); injection volume, 100 µL of 1 mg/mL peptide solution; flow rate, 1 mL/min; eluent A, 0.05% TFA in water; eluent B, 0.05% TFA in CH_3_CN; gradient, from 31% B to 41% B in 20 min, detection at 214 nm. The retention time result was 10.95 min (Appendix A). Experimental mass: 2519.3 Da, theoretical mass: 2519.8 Da (ESI-TOF, Mariner System 5220, Applied Biosystem, Perkin-Elmer, CA, USA) (Appendix A). A total amount of 28.38 mg of purified GBMP1a was obtained (9% yield considering the scale of synthesis).

### 2.3. S-X-HVP N-Terminal Conversion in an Alpha-oxo-Aldheyde

After being purified, S-X-HVP peptide was subjected to oxidation in order to provide the sequence with an alpha-oxo-aldehyde group at the N-terminal position [47]. The newly obtained peptide ald-X-HVP (OHC-CO-X-Phe-Arg-His-Arg-Asn-Arg-Lys-Gly-Tyr-NH_2_) was then ready to be anchored on chitosan. The reaction procedure is reported as follows. All purified S-X-HVP was dissolve in 0.04 M sodium phosphate buffer, to a final concentration of 1.9 mg/mL. After complete peptide dissolution, NaIO_4_ was added to reach 2.5 mM; the reaction time was 4 min at room temperature under magnetic stirring. The reaction mixture was filtered and purified through RP-HPLC at the following condition: column, Nova-Pak HR C_18_ (6 µm, 60 Å, 7.8 mm × 300 mm, Waters); eluent A, 0.05% TFA in MilliQ water; eluent B, 0.05% TFA in CH_3_CN; gradient, from 0% B to 8% B in 2 min, then from 8% B to 20% B in 24 min; flow rate, 4 mL/min; detection at 214 nm. The final peptide was over 99% pure. The identity of the peptide was ascertained through mass analysis (experimental mass: 1414.8 Da, theoretical mass: 1415.6 Da, ESI-TOF, Mariner System 5220, Applied Biosystem, Perkin-Elmer, CA, USA) (Appendix A). After oxidation, a total amount of 43 mg of ald-X-HVPa was obtained. Considering the synthesis scale, the yield of purified peptide with an appropriate aldehyde terminal is 24%. This yield, higher than the 9% of GBMP1a, proved that the strategy for the production of the alpha-oxo-aldheyde through the N-terminal conversion is more efficient than the direct synthesis of the aldheyde peptide on a specific support. Furthermore, the same HVP sequence was directly synthesized elsewhere on a H-Phe-H NovaSyn resin with a lower yield of 14% (considering the scale of synthesis) [45].

### 2.4. Scaffold Preparation 

#### 2.4.1. Chitosan Functionalization with ald-X-HVP and GBMP1a

The procedure followed to covalently bind each peptide to chitosan was reported elsewhere [46]. To obtain 43 mg of Chit-HVP, 40 mg of ald-X-HVP and 43.89 mg of Chitosan were used, while in the case of Chit-GBMP1a (30.88 mg of total final amount), only 19.95 mg of purified GBMP1a and 21.89 mg of chitosan were used.

#### 2.4.2. Chitosan Spongy Scaffold Preparation

Each scaffold was prepared by dissolving 0.84 mg of chitosan and 0.84 mg of Chit-HVP or Chit-GBMP1a in 0.252 mL of 0.2 M acetic acid. Two-hundred mg of solution was placed into each well of a 48-plate well plate. Samples were frozen with dry ice and lyophilized. After lyophilization, 2mL of EtOH was added to each sample and sonicated for 1 min, then left in EtOH for 2 min more. This procedure was repeated three times with EtOH and three times with 10× sodium phosphate buffer. Finally, all matrices were frozen with dry ice and lyophilized for 24 hr.

As a control, matrices composed of 1.68 mg of chitosan in 0.252 mL of 0.2 M acetic acid each were prepared (Chit). The procedure was the same as reported above.

### 2.5. Scaffold Characterization

#### 2.5.1. Fourier Transform Infrared Spectroscopy (FT-IR) Spectroscopy

Reflection-Absorption Infrared Spectroscopy (RAIRS) measurements were performed by means of a VECTOR 22 (Bruker) FT-IR interferometer operating in the wavenumber range 400–4000 cm^−1^, with a resolution of 1 cm^−1^, equipped with a Specac P/N 19,650 series monolayer/grazing angle accessory and with a DTGS detector. Spectra were recorded at incidence angles of 70° with respect to the normal to the sample surface.

#### 2.5.2. X-ray Photoelectron Spectroscopy (XPS)

A total of 0.62 mg of each sample was dissolved in 0.186 mL of 0.2 M acetic acid. Sixty µL of each sample was placed on a gold support ready for XPS analysis. XPS studies were performed using a homemade ultra-high vacuum (UHV) instrument equipped with a Mg kα X-ray source (hν = 1253.6 eV) and a five-channeltrons VG 150 mm mean radius hemispherical electron analyzer. C1s, Ca2p, O1s, B1s, Mg2p, Si2p, and N1s core level signals were recorded. The measured Binding Energies BE (±0.1 eV) were calibrated to the C1s signal of aliphatic-aromatic C-C carbons located at a BE = 285.0 eV [57]. Experimental spectra were analyzed using Gaussian curves as fitting functions. The same FWHMs (full widths at half maximum) were used for the different components in the same spectrum, and the same BEs were fixed for the identical components in the various spectra.

Atomic ratios (±10%) were calculated from peak areas using Scofield’s cross-section as sensitivity factors [58].

#### 2.5.3. Nuclear Magnetic Resonance analysis (NMR) of Chitosan, Chit-HVP, and Chit-GBMP1a in Solution

NMR samples were prepared by dissolving 1÷2.7 mg of chitosan, 2.7 mg of Chit-HVP, and 2.7 mg di Chit-GBMP1a in 700 μL of deuterated acetic acid 2 M in D_2_O. Acetic acid-d4 (99.9% isotopic purity) and D_2_O (99.9% isotopic purity) were purchased by Merck Life Science Srl, Milan, Italy. All the spectra were acquired at 298 K with a Bruker Avance NMR spectrometer operating at 700 MHz ^1^H Larmor frequency. ^1^H spectra were acquired with 16–32 scans, inter-scan delay of 10 s, and water suppression with z-gradients. The residual methyl resonance of deuterated acetic acid (CHD_2_-COOD) at 2.03 ppm was used for spectra reference.

#### 2.5.4. Scanning Electron Microscope Analysis (SEM)

Scanning electron microscope (SEM, Mod JSM-6490 Jeol, Peabody, MA, USA) analysis was performed in order to evaluate the surface topography of Chit, Chit-HVP, and Chit-GBMP1a matrices. All samples were sputter coated with gold (Quorum sputter coater model Q150R E, Quorum Technologies, Laughton, UK). All analyses were performed using a tungsten thermionic emission SEM system suitable for high-vacuum operations, equipped with an energy dispersive X-ray detector (EDX), model IXRF Systems 500 (Jeol, Peabody, MA, USA).

### 2.6. Biological Assays

#### 2.6.1. Cell Culture

Human bone cells were obtained from explants of cortical mandible bone collected during a routine surgical procedure from a 32-year old healthy adult man. The local ethics committee approved the study, and informed consent was obtained from the patient. Bone fragments were cultured in Dulbecco’s modified eagle medium (D-MEM)/F12 (1:1) supplemented with 20% *v*/*v* fetal bovine serum, 1% *v*/*v* sodium pyruvate, 1% *v*/*v* nonessential amino acids, 1% *v*/*v* antibiotic–antimycotic solution, and 1 U/mL insulin (complete medium; all reagents were provided by Gibco, Invitrogen, Milan, Italy) until cells migrated from tissue. At confluence, bone fragments were removed and cells were detached by Trypsin-ethylenediaminetetraacetic acid (EDTA, Gibco). Then, cells were cultured in complete medium supplemented with 50 μg/mL ascorbic acid, 10 nM dexamethasone, and 10 mM β-glycerophosphate (all provided by Sigma, Milan, Italy). After a 10-day culture, cells were seeded on 12-well culture plates and 24 h later subjected to alkaline phosphatase (ALP) activity and von Kossa staining. ALP activity was assessed in cell monolayers using the Alkaline Phosphatase Assay Colorimetric Kit (abcam; Milan, Italy), following the instructions of the vendor. Briefly, cells were added of p-nitrophenyl phosphate (pNPP), the phosphatase substrate of ALP. Reactions were stopped 60 min later by acidification, and optical densities were recorded using a microplate reader (Tecan) at 405 nm. Deposition of mineral salts was investigated by von Kossa staining in cells cultured on a glass coverslip. Cells were fixed in paraformaldehyde (PFA) 4% w/vol, washed in distilled water, and then incubated for 20 min in 5% w/vol aqueous silver nitrate solution under UV exposure. Samples were then washed and incubated in 2% w/vol sodium thiosulphate for 2 min. Cells were counterstained with 1% neutral red and visualized. At the time of the experiments, cells were detached using Trypsin-EDTA, washed, and resuspended in complete supplemented medium at a density of 1 × 10^5^ cells/100 μL. Cells were placed on the surface of matrices (1 cm^3^) for five minutes to allow the matrices to adsorb the volume. Then, the matrices were completed dipped in culture medium and incubated at 37 °C, 5% CO_2_. When required, culture media were renewed every two days; changes in pH of the media or evaporation of the media were periodically inspected.

#### 2.6.2. Cell Viability Assay

Viability of cultured cells was assessed using the MTT (3-(4,5-dimethylthia- zole-2-yl)-2,5-diphenyl tetrazoliumbromide; Merck, Kenilworth, NJ, USA). Cultured human osteoblast cells were seeded on matrices and incubated at 37 °C for 2 h, a time previously reported, ensuring the optimal adhesion of h-osteoblasts to functionalized surfaces [9]. At the end of incubation, matrices were rinsed three times with phosphate buffered saline (PBS) to remove non-adherent cells. Then, matrices were incubated with MTT (5 mg/mL) at 37 °C for 4 h under agitation. The reaction was stopped by adding 0.01 N HCl in 10% v/v sodium dodecyl sulfate (SDS). To quantify adherent cells, we set a standard curve obtained for each experiment by seeding a known number of osteoblasts. The absorbance of cells lysates was recorded at 620 nm.

#### 2.6.3. Cell Proliferation Assay

The proliferation of osteoblasts cultured on matrices was assessed using carboxyfluorescein diacetate succinimidyl ester (CFSE), a cell-permeable fluorescent probe equally partitioned among daughter cells. Cells were first incubated with CFSE 25 µM at 37 °C for 10 min in pre-warmed PBS containing 0.1% *v*/*v* bovine serum albumin (BSA). The reaction was stopped by adding 5 volumes of ice-cold culture media. Cells were then washed, counted using Trypan blue, and seeded on matrices as described above. Four days later, matrices were collected, washed in PBS, and incubated for 5 min in trypsine-EDTA under gentle agitation. Cell proliferation was assessed using a BD fluorescence-activated cell sorting (FACS)-Calibur flow cytometer by evaluating the percentage of CFSE-positive cells in 10,000 events.

#### 2.6.4. Quantitative Real-Time Polymerase Chain Reaction

Specific mRNA transcript levels coding Runt-related transcription factor 2 (*RUNX2*), Vitronectin (*VTN*), and Osteopontin (*SPP1*) were quantified in osteoblast cells cultured for 24 h on matrices. At the end of incubation, matrices were washed in PBS and incubated in SV Total RNA Isolation System kit (Promega, Milan, Italy). Total RNA was extracted, and contaminating DNA was removed by DNase I digestion. cDNA synthesis and subsequent polymerization were performed in one step using the iTaq Universal SYBR Green One-Step Kit (Bio-Rad). The reaction mixture contained 200 nM forward primer, 200 nM reverse primer, iTaq universal SyBR Green reaction mix, iScript reverse transcriptase, and 200 ng total RNA. Real time polymerase chain reaction (PCR) was performed using ABI PRISM 7700 Sequence Detection System (Applied Biosystems, Foster City, CA, USA). Data were normalized using glyceraldehyde-3-phosphate dehydrogenase (*GAPDH*) as the reference gene. Target and reference genes were amplified with efficiencies near 100%. Oligonucleotides used for PCR are listed in Table 1. 

#### 2.6.5. Calcium Assay

Intra- or extra-cellular formation and deposition of calcium phosphate crystals are considered hallmarks of osteoblast differentiation [59]. As previously reported, calcium content is undetectable at 2 days of culture, peaks at 7 days, and decreases at 14 days [60]. Therefore, this study assessed calcium levels in osteoblasts cultured on matrices for 7 days. At the end of incubation, matrices were washed in PBS and cells detached as described above. Cells were fixed using 10% PFA for 30 min and then stained with 40 mM Alizarin red (pH 4.2) for 40 min in the dark at room temperature. The cells were incubated at −20 °C for 30 min, and then lysed in acetic acid 10% *v*/*v*. The samples were incubated at 85 °C for 10 min and centrifuged. The pH of the supernatants was neutralized before reading the absorbance of Alizarin red at 405 nm using a microplate reading (Tecan) [61].

### 2.7. Statistical Analysis

Biological data are reported as mean ± standard error of two independent experiments, each performed in triplicate. Statistical analysis was performed using the one-way ANOVA test followed by Bonferroni’s multicomparison test, using Graph-Pad Prism 3.03, and *p*-values < 0.05 were considered statistically significant.

## 3. Results

### 3.1. Scaffold Characterization

#### 3.1.1. FT-IR

Chit-HVP and Chit-GBMP1a infrared spectra are reported in Figure 1; the spectrum of pristine chitosan is also reported for comparison.

The main features in the chitosan spectrum (black line) are the broad band located at 3550 cm^−1^ and related to O-H stretching vibrations and the peak located at 1080 cm^−^^1^ due to C-O stretching vibrations and labelled ν_C-O_ in Figure 1. Since the hydrolysis of the amide functions present in the chitin molecules is seldom complete, peaks related to C=O stretching and N-H bending vibrations of residual amides are evident in the spectrum, located at 1650 cm^−1^ (ν_C=O_) and 1560 cm^−1^ (labelled δ_N-H_ ), respectively [62].

The spectra of the two peptide-functionalized samples, Chit-HVP (blue) and Chit-GBMP1a (red), are very similar to the spectrum of pristine chitosan, with the same peaks in the same position. However, the intensity ratio between the amide-related peaks (at 1650 and 1560 cm^−1^) and the C-O stretching peak of chitosan (at 1080 cm^−1^) is higher in the peptide-functionalized samples compared to pristine chitosan. This effect is proof of the successful peptide immobilization, since peptide bonds also contribute to the amide bands at 1650 and 1560 cm^−1^.

#### 3.1.2. XPS Spectroscopy

For all samples, XPS measurements of core levels C1, N1, and O1 were performed. The C1s and N1s spectra of Chit-HVP, Chit-GBMP1a, and Chit are shown in Figure 2.

For all the investigated samples, the spectrum of C1s signal can be resolved by four-component curve-fitting analysis associated with carbon atoms that are not chemically equivalent. Peak 1 (BE = 285.0 eV) is attributed to aliphatic carbons (C-C) of the side chains and partially to carbon contamination, which cannot be completely eliminated even in UHV (ultra-high vacuum); peak 2 (286.5 eV) correlates to the C-N carbons of the peptide chain, with the contribution of the C-N carbons of the lysine groups; and peak 3 (288.0 eV) is due to peptide carbons O = C-N. Peak 4 (290.3 eV) is due to carbons with HN = C-NH_2_ in arginine [57,63,64,65]. Peak 5, located at 282 eV only in the spectrum of sample Chit-HVP, is probably a spurious signal, possibly due to weak surface contamination.

The O1s spectra (not shown) consist of a single peak located at 532.4 eV and related to organic oxygens [57,66].

The N1s spectrum consists of a single peak at 399.17 eV due to N-C type nitrogen atoms of the arginine group and peptide nitrogen; a small contribution at about 401.0 eV and due to protonated nitrogens is evidenced only for pristine chitosan [65,66].

The contribution of the C-O carbons of the chitosan is located at 286.6 eV and overlaps with the signal of the C-N carbons. In a similar way, it is possible to attribute the bands, documented by the deconvolution of the Chit-GBMP1a spectra, to the groups that make up this functionalized polymer.

Analysis of the XPS data confirmed that the anchoring of both peptides to chitosan was successful. In particular, the increased N/C ratio for Chit-HVP and Chit-GBMP1a matrices compared to the control (Chit) is particularly significant, since the peptide molecules have a higher nitrogen content (Table 2).

The increase of N/C was due to the anchoring of the peptides, as the N/C ratio in chitosan between 1/6 (0.16) and 1/8 (0.125), while the N/C ratio for X-HVP is 0.3 and for GMBP1a is 0.267.

#### 3.1.3. NMR Analysis of Chitosan, Chit-HVP, and Chit-GBMP1a in Solution

Chit, Chit-HVP, and Chit-GBMP1a were analysed in deuterated acetic acid 2 M in D_2_O using a spectrometer operating at 700 MHz. The ^1^H spectrum of Chit (Figure 3a) is in agreement with the literature data and shows the resonances of the H3–H6 protons of glucosamine (GlcN) and acetyl-glucosamine (GlcNAc) in the range of 4.0–3.5 ppm, and that of the H2 proton of GlcN at 3.15 ppm. The anomeric H1 resonances of both monomers, typically at ~4.9 ppm, overlap the residual water signal and are then cancelled by the water suppression procedure. The spectrum also shows the methyl protons of the acetyl group (GlcNAc) at 2.05 and 2.04 ppm, where the last and larger line overlaps the first line of the quintuplet associated to the resonance of the residual CHD_2_COOD (see the inset in Figure 3a). The two signals for the methyl group are reasonably due to differences about the aggregation state. The acetylation degree (DA) of soluble chitosan was evaluated according to the literature [66] by the ratio between the integral (I_CH3_) of the methyl group at 2.05 and 2.04 ppm and that of the H2–H6 protons according to the formula:$$\%DA=\frac{\left(2\times I_{CH3}\right)\;}{\left(I_{H2-H6}\right)}\times 100$$

By using the data reported in Figure 3a, we found that our chitosan sample shows an acetylation degree of about 24%:$$\%DA=\frac{\left(2\times 1.80\right)}{\left(10.0+5.2\right)}=0.237\%$$

The spectrum of Chit-HVP is shown in Figure 3b. All the resonances due to chitosan overlap with signals from the peptide. To estimate the functionalization of chitosan by HVP, R_f_, we used the ratio between the integral of one peptide proton (I_pept. H_) and one chitosan proton (I_chit. H_). By considering the integrals of the region 7.30–6.75 ppm (10 HVP aromatic protons), we found I_pept. H_ = 1.0. To determine I_chit.H_, the integrals of 4.0–3.5 and 3.16 ppm regions, containing six protons of chitosan and twelve of peptide, were considered. I_chit.H_ was obtained by subtracting from the total integral (46.1, see Figure 3b) the value corresponding to the peptide contribution, 10, and then dividing the resulting value by 6. Determined in this way, R_f_ = I_pept.H_/I_chit.H_ is equal to 0.17. This suggests that our chitosan sample is HVP functionalized by about 17%.

The degree of chitosan functionalization by GBMP1a was estimated in an analogous way. For the Chito-GBMP1a spectrum (Figure 3c), the value of I_pept.H_ was obtained by the integral of the aromatic region at 7.35–7.20 ppm (12 peptide protons), while I_chit.H_ was obtained by subtracting from the total integral, 57.6, the value corresponding to 16 peptide protons, and then dividing by 6. The ratio I_pept.H_/I_chit.H_ = 0.833/7.38 returns a chitosan functionalization by GBMP1a of about 11%.

The NMR analysis shows that the plain chitosan sample is acetylated by 24%. This means that 76% of chitosan chains are GlcN units and can react with the peptide. By considering the ratios 0.17/0.76 and 0.11/0.76, we found that the reaction yield is 22% and 14% for HVP and GBMP1a, respectively.

The NMR analysis of the Chit-peptides was performed under identical solvent and temperature conditions. In these conditions, the chitosan and Chit-HVP produce perfectly transparent solutions, whilst Chit-GBMP1a shows a weak but appreciable turbidity. Hence, the degree of derivatization of chitosan with the two peptides refers to the soluble components of the adducts.

#### 3.1.4. SEM Analysis

Chit, Chit-HVP, and Chit-GBMP1a SEM images are shown in Figure 4.

From the images of the Chit and Chit-HVP spongy matrices, it was possible to observe the typical open porous structure. A slightly different morphology with softer structures was observed for Chit-GBMP1a functionalized foam matrix. This could be attributed to the different nature of the two peptides: HVP peptide is highly basic and is positively charged as plain chitosan, whilst GBMP1a peptide is lightly charged (+2) and much less hydrophilic.

### 3.2. Biological Assays

#### 3.2.1. Functionalized Matrices Are Nontoxic for Osteoblast Cells and Support Cell Proliferation

The purity of cultures of osteoblast cells was greater than 96%, as confirmed by ALP activity and von Kossa staining (data not shown). Chit-HVP and Chit-GBMP1a did not report toxic effects in human osteoblast cells and allow cell attachment. Indeed, as reported in Figure 5a, after two hours in culture, we recovered more cells from chitosan functionalized matrices compared with chitosan matrices alone. Moreover, Chit-HVP held significantly more cells (1.6 fold, *p* < 0.05) than Chit-GBMP1a (Figure 5a).

We next measured cell proliferation in osteoblasts cultured for 4 days on different matrices. As reported in Figure 5b, functionalization greatly increased cell proliferation. Indeed, cells cultured on Chit-HVP reported an increase of about 14 folds compared with Chit alone. At the same time, Chit-GBMP1a increased cell proliferation by about 12.3 fold compared with Chit alone. The effects on cell proliferation did not statistically differ in cells cultured on Chit-HVP or Chit-GBMP1a.

#### 3.2.2. Functionalized Matrices Induce Differentiation in Cultured Osteoblast Cells

To assess the ability of functionalized matrices in supporting osteoblast cells differentiation, cells were cultured for 7 days and then subjected to Alizarin staining to evaluate calcium salts deposition.

On the seventh day of the culture, we observed that calcium deposition was greater in cells cultured on Chit-GBMP1a than cells cultured on Chit and Chit-HVP (Figure 6a).

Chit-HVP induced salt deposition compared with Chit alone, but the effect was less important than cells cultured on Chit-GBMP1a, demonstrating this functionalization’s ability to guide and maintain the osteoblastic phenotype in primary human cells. Indeed, calcium deposition is seven times higher in cells cultured on Chit-GBMP1a than in control (Chit), while Chit-HVP matrices increased the control value by approximately 0.5 fold. Therefore, even if Chit-HVP promoted the initial osteoblasts’ adhesion (Figure 5a), Chit-GBMP1a guides the differentiation of cells and the salt deposition in human osteoblast primary cells.

The ability of bioactive peptides in supporting cell differentiation was further confirmed by measuring the mRNA transcript levels of three crucial osteoblast genes. As reported in Figure 6b, both Chit-HVP and Chit-GBMP1a significantly increased gene expression compared with Chit alone. However, Chit-GBMP1a dramatically increased the expression of *SPP1*, *VTN*, and especially *RUNX2* genes compared with Chit-HVP.

## 4. Discussion

Nowadays, attention towards the use of natural materials that induce positive responses from the host is greatly increasing [67]. In this scenario, chitosan, with its low-cost and economic nature, has been widely studied and extensively used in biopharmaceutical and biomedical applications. In the field of bone tissue engineering, especially, the ability of chitosan to guide osteogenesis has been proven. Furthermore, incorporating bioactive molecules into chitosan-based scaffolds accelerates new bone regeneration and enhances neovascularization in vivo [18,19]. For these reasons, the importance of identifying a simple and rapid way to covalently bind bioactive signals to scaffold matrices is evident.

In this work, we proposed a very easy and cheap procedure to covalently graft HVP and GBMP1a bioactive peptides to chitosan through a reaction involving an aldehyde group of the peptide and the amino groups of the polymer [45,46]. The oxidation of the Serine led to a higher yield of HVP aldehyde than GBMP1a aldehyde (24% vs. 9% considering the scale of synthesis).

FT-IR and XPS characterization of the matrices demonstrated the efficacy of the functionalization. The peptides’ presence is proven by the increased N/C ratio in functionalized scaffolds (0.15 for Chit-HVP and 0.10 for Chit-GBMP1a) with respect to unfunctionalized chitosan (0.07). Furthermore, NMR analysis confirmed an 11% and 17% functionalization of chitosan with GBMP1a and X-HVP, respectively. The substitution degree (DS) results were lower than those reported in Batista M. et al. [68]—as expected, considering the different peptides sequence length.

SEM imaging showed the porous morphology of the 3D spongy matrices of chitosan and functionalized chitosan, which are useful for BTE scaffolds.

Chitosan and the functionalized matrices did not report toxicity in human primary osteoblast cells. Indeed, chitosan enrichment with X-HVP significantly increased cell adhesion at 2 h compared with chitosan functionalized with GBMP1a peptide (Figure 5a), confirming the critical role of HVP peptide in anchoring osteoblast cells, as previously reported by our group [45,54]. However, the importance of functionalization with HVP appears time-limited, as following 4 days in culture cell proliferation, cell growth was comparable in Chit-HVP and in Chit-GBMP1a functionalized matrices (Figure 5b). Our data demonstrate the biocompatibility of both peptides and their ability to support cell proliferation, a critical skill of biomaterial during tissue regeneration. During bone tissue healing, the cell proliferation is not sufficient and needs to be accompanied by cell differentiation to eventually ensure the formation of functional tissue. GBMP1a peptide proved to be a very powerful osteoblasts differentiation factor not only in the soluble form [56], but even when conjugated to chitosan (Figure 6). In particular, Chit-GBMP1a reported a dramatic enhancement in calcium deposition and expression of genes involved in osteoblast differentiation. In particular, cells cultured on Chit-GBMP1a showed increased mRNA expression levels of vitronectin (*VTN*) and osteopontin (*SPP1*) coding extracellular matrix proteins essential in osteogenic differentiation [69,70]. The robust induction of *RUNX2* mRNA transcripts in cells cultured on GBMP1a chitosan matrices (Figure 6b) could justify the cell proliferation recorded in these samples despite the lower number of cells recovered following 2 h of culture compared to Chit-HVP (Figure 5a) [71].

In the future, Chit-GBMP1a scaffolds will be tested as a support to guide mesenchymal stem cells adhesion, growth, proliferation, and differentiation, also in association with Chit-HVP, with the hope of a synergic effect.

## Figures and Tables

**Figure 1 nanomaterials-11-02784-f001:**
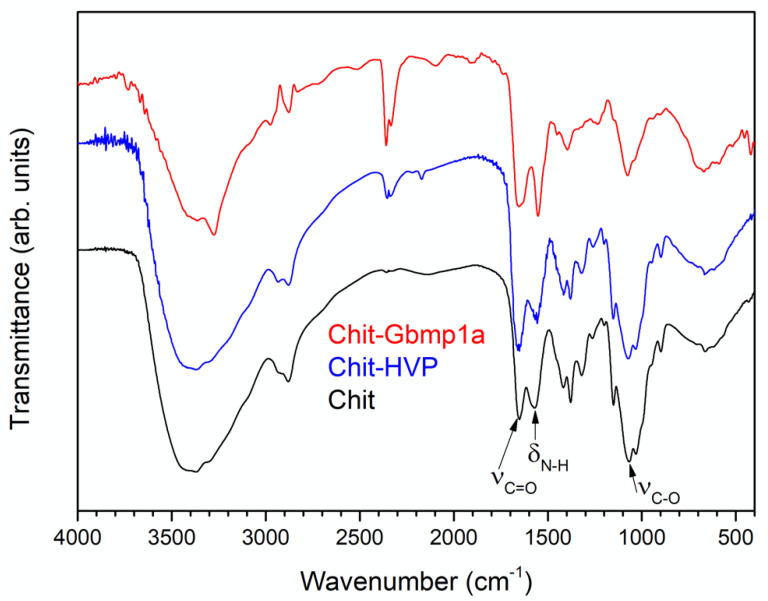
FT-IR spectra of Chit (black), Chit-HVP (blue), and Chit-GBMP1a (red).

**Figure 2 nanomaterials-11-02784-f002:**
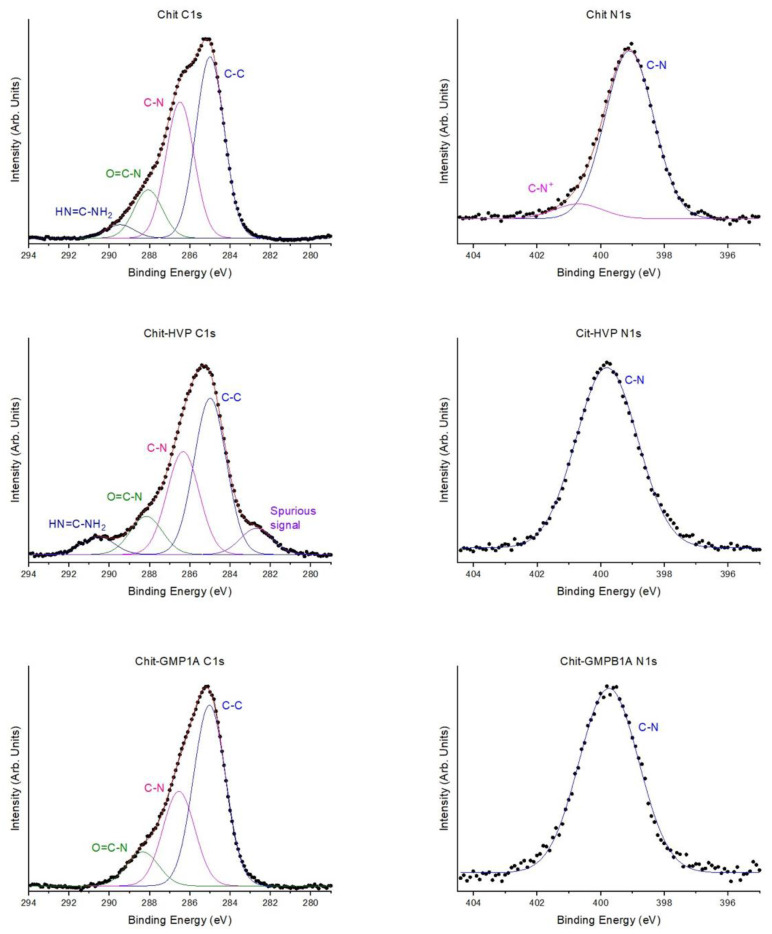
C1s and N1s experimental spectra of Chit, Chit-HVP, and Chit-GBMP1a deposited on gold with curve-fitting analysis. Markers represent experimental points; lines represent fitting components and calculated spectra.

**Figure 3 nanomaterials-11-02784-f003:**
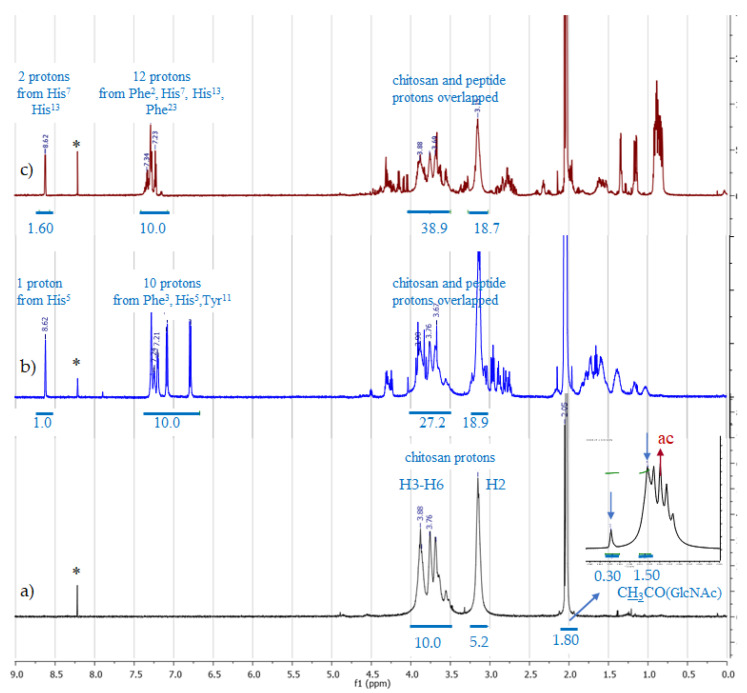
^1^H-NMR spectra of Chit (**a**), Chit-HVP (**b**), and Chit-GBMP1a (**c**) in deuterated acetic acid (ac) 2 M in D_2_O. The inset in (**a**) shows the expansion of the region 2.08–2.00 ppm, where the CH_3_CO group of GlcNAc shows two resonances, at 2.05 and 2.04 ppm (blue arrows). The 2.04 resonance overlaps the first line of the quintuplet due to CHD_2_COOD (ac). Blue lines and numbers under the peaks represent the integrated regions and the corresponding normalized integrals, respectively. * = impurity.

**Figure 4 nanomaterials-11-02784-f004:**
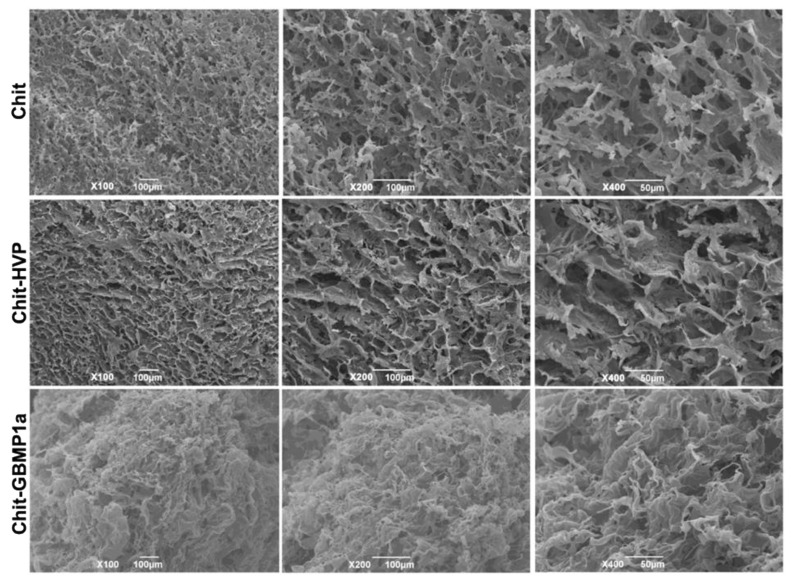
SEM images showing characteristic matrices morphology. Images were taken at three different enlargements, X100, X200, and X400, for Chit, Chit-HVP, and Chit-GBMP1a scaffolds.

**Figure 5 nanomaterials-11-02784-f005:**
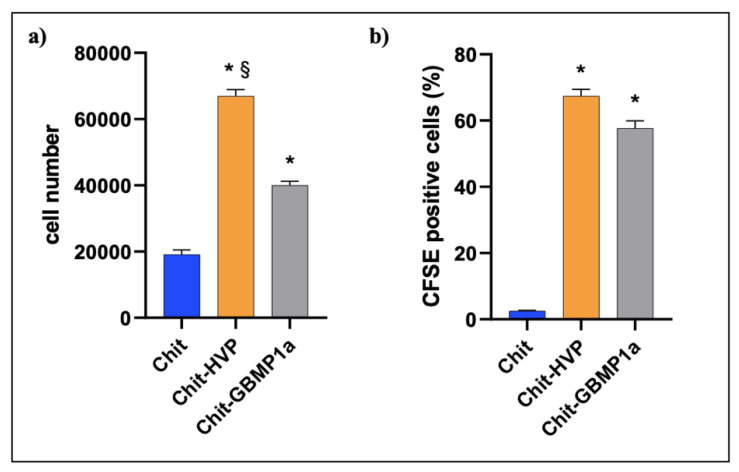
Cell viability assays. (**a**) Osteoblast adhesion at 2 h and (**b**) osteoblast proliferation at 4 days of culture on Chit, Chit-HVP, and Chit-GBMP1a matrices. * *p*-value < 0.05 vs. Chit; ^§^
*p*-value < 0.05 vs. Chit-GBMP1a.

**Figure 6 nanomaterials-11-02784-f006:**
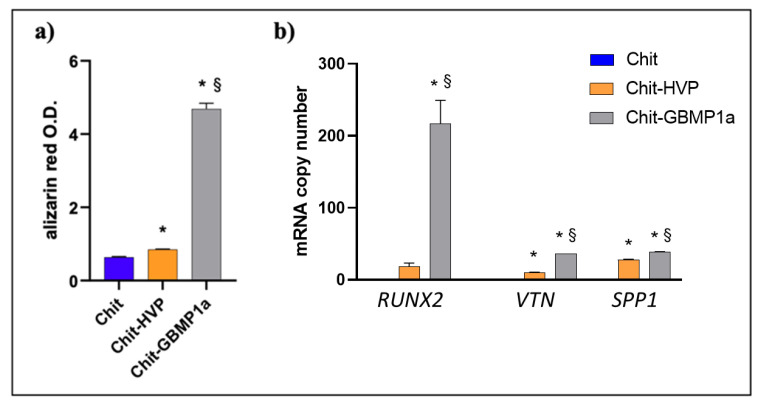
Calcium salt deposition and gene expression assays. (**a**) Levels of calcium were evaluated in osteoblasts cultured for 7 days in differently functionalized chitosan matrices. O.D.: optical density. (**b**) Human *Runx2 Vtn* and *Spp1* mRNA-specific transcript levels were evaluated by quantitative RT-PCR on cells cultured on Chit, Chit-HVP, and Chit-GBMP1a for 24 hr. * *p*-value < 0.05 vs. Chit; ^§^
*p*-value < 0.05 vs. Chit-HVP.

**Table 1 nanomaterials-11-02784-t001:** Oligonucleotides sequences of the genes evaluated by quantitative PCR.

Gene	Sequence
*GAPDH*	Fw: 5′-agtgccagcctcgtcccgta-3′ Rv: 5′-caggcgcccaatacggccaa-3′
*RUNX2*	Fw: 5′-cagtgacaccatgtcagcaa-3′ Rv: 5′-gctcacgtcgctcattttg-3′
*VTN*	Fw: 5′- ggaggacatcttcgagcttct-3′ Rv: 5′- gctaatgaactggggctgtc-3′
*SPP1*	Fw: 5′-aagtttcgcagacctgacatc-3′ Rv: 5′-ggctgtcccaatcagaagg-3′

**Table 2 nanomaterials-11-02784-t002:** N/C ratio measured for Chit, Chit-HVP and Chit-GBMP1a.

Sample	N/C Ratio
Chit	0.07
Chit-HVP	0.15
Chit-GBMP1a	0.10

## Data Availability

The data presented in this study are available within this article and in Appendix A.

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
