# Peer review of "Chitosan Covalently Functionalized with Peptides Mapped on Vitronectin and BMP-2 for Bone Tissue Engineering"

_nanomaterials, 2021, doi:10.3390/nano11112784_

Round 1

Reviewer 1 Report

This manuscript described the development of chitosan scaffold functionalized with two peptides and test the cellular response.

  1. abstract: please add the conclusion description in the abstract.
  2.  On page 3, why selecting HVP and BMP-2 as the test peptides rather than other peptides (ALP......) or integrins. Does the author consider the synergistic efficiency of  HVP and BMP-2 together? 
  3.   In Figure1, please mark the special findings (for example amide-related peaks) in figure 1.
  4. In Figure2 and 3, experimental data and important findings are difficult to be understood. Please mark the important findings in the figures.
  5. In figure 4, why the SEM morphology and porosity characteristics of the chit-GBMP1a is different from the other groups?
  6. For osteoblastic differentiation, do authors provide the gross or microscopic images for calcium deposition confirmation?
  7.   Furthermore, how to confirm the best amount of peptides suitable for bone tissue engineering?   

Author Response

Point-by-point response to Reviewer's comment

Reviewer 1

This manuscript described the development of chitosan scaffold functionalized with two peptides and test the cellular response.

  1. abstract: please add the conclusion description in the abstract.

Answer: The authors would like to thank the reviewer for the suggestion. The conclusions of the study are already stated in the last three lines of the abstract: “Chit-HVP and Chit-GBMP1a porous scaffolds promoted human osteoblasts adhesion, proliferation, calcium deposition, and gene expression of three crucial osteoblast proteins. In particular, Chit-HVP highly promoted adhesion and proliferation of osteoblasts, while Chit-GBMP1a guided cell differentiation towards osteoblastic phenotype.”

  1. On page 3, why selecting HVP and BMP-2 as the test peptides rather than other peptides (ALP......) or integrins. Does the author consider the synergistic efficiency of HVP and BMP-2 together?

Answer: In this study we selected two bioactive peptides for covalent functionalization of chitosan as we wanted to improve osteoblast adhesion and differentiation. For this reason, to enhance chitosan ability to sustain osteoblast adhesion we selected HVP since the ability of this peptide to improve osteoblast adhesion has already been demonstrated [DOI:10.1016/j.actbio.2012.12.018] and, unlike RGD or other adhesive sequence, it is osteoblast-specific. Furthermore, to guide cell differentiation we decided to graft GBMP1a, which capacity to induce cell differentiation into osteoblast was already assessed [DOI:10.1016/j.bbagen.2017.07.001]. The results of this preliminary study open up the possibilities of covalently and selectively binding whatever bioactive motif to promote a particular cellular response of interest.

As stated in the last three lines of the discussion (lines 520-522), in this study the synergistic effect of the two peptides was not taken into account, but it will be tested in future experiments. We thank the reviewer for the suggestion.

  1. In Figure1, please mark the special findings (for example amide-related peaks) in figure 1.

Answer: The most important peaks were labelled in the figure as requested. Small changes were made in lines 333-336 pag 8:

“… and labelled nC-O in figure 1. Since the hydrolysis of the amide functions present in the chitin molecules is seldom complete, peaks related to C=O stretching and N-H bending vibrations of residual amides are evident in the spectrum, located at 1650 cm-1 (nC=O) and 1560 cm-1 (labelled dN-H) respectively”

  1. In Figure2 and 3, experimental data and important findings are difficult to be understood. Please mark the important findings in the figures.

Answer: Figure 2 and 3 were changed according to the reviewer’s request. A sentence was added in the figure 2 caption:

“Markers represent experimental points, lines fitting components and calculated spectra.”

To make the results obtained from NMR analysis clearer the related paragraph 3.1.3 was completely rewritten, and the method used for the obtainment of the substitution degree of Chit-HVP and Chit-GBMP1a was described more accurately. Accordingly, line 598-599 were changed as follows:

“Furthermore, NMR analysis confirmed a 11% and 17% functionalization of Chitosan with GBMP1a and X-HVP, respectively.”

  1. In figure 4, why the SEM morphology and porosity characteristics of the chit-GBMP1a is different from the other groups?

Answer: We thank the reviewer for the question. SEM morphology is slightly different for Chit-GBMP1a scaffolds probably due to the different charge of GBMP1a peptide. In fact, GBMP1a, is less positively charged than HVP and this results in a less hydrophilicity. This could cause a different water retention during the freeze-drying process that leads to a different scaffold morphology and porosity.

  1. For osteoblastic differentiation, do authors provide the gross or microscopic images for calcium deposition confirmation?

Answer: Osteoblastic differentiation was assessed by alizarin staining. Then cells were lysed and optical density was recorded using microplate spectrophotometer. The protocol allowed us to obtained relative analytical determination of salts deposited by cells. We agree with the Reviewer that alizarin stained cells could be observed also by microscopic analysis. Unfortunately, in this study the scaffolds were not suitable for microscopic investigation.

  1. Furthermore, how to confirm the best amount of peptides suitable for bone tissue engineering?

Answer: The authors would like to thank the reviewer for the interesting question. In this study we performed a functionalization of chitosan using only one working peptide concentration, but in the next future different grade functionalization could be achieved modifying peptide solution concentration in order to maximize osteoblast adhesion or differentiation.

Reviewer 2 Report

This work investigated the potential application of chitosan covalently functionalized with peptides mapped on vitronectin and BMP-2 for bone tissue engineering. The authors performed numerous physical property assessments, including biological evaluations.

  1. This journal is about nanomaterials. However, there is no mention of nano-level materials or structures in the authors' text. The authors should explain why their paper should be published in this journal.
  2. In 2.6.1, the authors write that they obtained “human bone cells”. However, the reviewer thinks they selected osteoblasts from the protocol, so it would be better to describe them accurately. The authors then state that the purity of the osteoblasts was confirmed by ALPL activity and von Kossa staining. They need to describe more precisely how they achieved purity by these two methods.
  3. The authors need to include how many times each biological assay was performed.
  4. Although the authors state in the Introduction and at the beginning of the Discussion that chitosan is an excellent biomaterial, the results they show for chitosan in Figures 5 and 6 call this into question. In particular, the cell proliferation results shown in Figure 5b show little proliferation with chitosan. Is chitosan really a good scaffold for bone tissue?
  5. Do the results for calcium salt deposition shown in Figure 6a take into account differences in cell proliferative potential? Since the reviewer does not know the number of cells on day 7, it is difficult to assess the cell differentiation status of individual osteoblasts from this result.
  6. The reviewer doesn't think the value of the y-axis in Figure 6b shows ΔΔCt. Please correct this accordingly. As for the RNA used in this experiment, Figure 5 shows that the number of cells attached to chitosan is less than 20,000 and those cells hardly proliferate. In this case, wouldn't the kit used by the authors be able to extract 200 ng of total RNA for the cDNA synthesis described in the Materials and Methods?

Minor points:

  1. What are “h-osteoblasts” in line 25?
  2. What are “GAGs” in line 25?
  3. The second 2.4.1 appears in line 211.
  4. The chapter number 2.5 appears four times in the Material and Methods.

Round 2

Reviewer 1 Report

Agree 

Reviewer 2 Report

I agree to publish this manuscript.